# Plasma Enhanced-Chemical Vapor Deposition of 2-Isopropenyl-2-Oxazoline to Promote the Adhesion between a Polyethylene Terephthalate Monofilament and the Rubber in a Tire

Carlo Maria Gaifami [1,*], Stefano Zanini [2], Luca Zoia [3] and Claudia Riccardi [2]

1   Dipartimento di Scienze dei Materiali, Università degli Studi di Milano-Bicocca, Via Roberto Cozzi, 55, I-20126 Milano, Italy

2   Dipartimento di Fisica, Università degli Studi di Milano-Bicocca, Piazza della Scienza, 3, I-20126 Milano, Italy; stefano.zanini@unimib.it (S.Z.); claudia.riccardi@unimib.it (C.R.)

3   Dipartimento di Scienze dell'Ambiente e della Terra, Università degli Studi di Milano-Bicocca, Piazza della Scienza, 1, I-20126 Milano, Italy; luca.zoia@unimib.it

*   Correspondence: c.gaifami@campus.unimib.it

**Abstract:** A Plasma-Enhanced Chemical Vapor Deposition was chosen in order to deposit an organic thin film on polyethylene terephthalate monofilament to increase its adhesion with the rubber compound in a tire. The aim of the work is to find an alternative "green" method to the classical chemical dipping with Resorcinol Formaldehyde Latex: plasma treatments are environmentally friendly and easy to use. 2-isoprepenyl-2-oxazoline (2-iox) was employed as precursor and the treatments were performed in a vacuum system, both in a continuous regime and a pulsed regime. Initially, the coatings were deposited on polyethylene terephthalate sheets to study the wettability (by the measurement of contact angle) and the thickness (by profilometer) of the plasma polymer. The chemical characterization was investigated by Infrared and X-ray Photoelectron spectroscopies. Finally, the adhesion of the polyethylene terephthalate sheets was measured by Peel Test, using the coating as adhesive and as a pre-dip. The measurement of the peel force made it possible to optimize the plasma parameters that were applied on the monofilament. The adhesion was estimated by the measure of the extraction force and the evaluation of the coverage compared with those of the classical chemical treatment Resorcinol Formaldehyde Latex.

**Keywords:** plasma enhanced-chemical vapor deposition (PE-CVD); polymers; fibers; adhesion; tire

## 1. Introduction

Tire is a composite material, where man-made organic fibers are employed to reinforce the rubber compound; in the tire industry, the most used organic fibers are Rayon (regenerated cellulose), Polyamide 6/6.6, Polyethylene terephthalate (PET) and Aramid (aromatic polyamide). The adhesion between the organic textile reinforcing materials and the rubber is a crucial aspect to guarantee the tire integrity and durability: from the 1940s to the present, the adhesion was made possible by the Resorcinol Formaldehyde Latex (RFL) dipping [1–3]. The RFL is efficient as an adhesive due to the crosslinking of the system with the compound chains [2]. PET and Aramid are less reactive with the RFL than the other man-made organic fibers; for this reason, a pre-bath is necessary to activate the fiber. The pre-dip introduced by DuPont is the most used in the tire industries; the chemical is prepared by water, miscible epoxy, and blocked isocyanate [4]. In recent years, researchers have been focused on finding new, alternative methods to promote the adhesion between a textile and rubber, because the presence of the formaldehyde and resorcinol has negative environmental impacts [5,6]. Different approaches from different authors were recently explored: one strategy was the preparation of formaldehyde-free adhesive using acrylic

resin, another strategy was chemical etching with bromine or the insertion of blocked isocyanate in the compound to promote adhesion [7–9]. In this view, plasma technology is an increasingly promising technique. Cold plasma could be created in a reactor by the application of an electric potential difference to a gas [10,11]. Thanks to this environmentally friendly technique, it is possible to change the morphology of the surface or to deposit a thin film on different substrates using limited doses of reagents [12–15]. If the gas used is an organic compound, the coating can be defined as Plasma Polymer (PPOL). PPOLs are new materials, they differ from a classical polymer by the structural unit, the length of chains and the degree of the crosslinking, which is usually higher [16]. Plasma treatments were recently used to promote the adhesion between a textile material and a rubber compound. "The General Tire & Rubber Company" patented a plasma activation followed by a grafting with vinyl pyridine before dipping in the RFL, to improve the adhesion of Aramid cord [11]. The same strategy was explored by Goodyear: plasma activation was followed by the polymerization of $CS_2$ to develop a network with the compound [17]. Another plasma approach was explored by Mzabi et al. and based on PE-CVD (Plasma-Enhanced Chemical Vapor) using organic molecules (i.e., maleic acid) to deposit a PPOL [18]. In this work, the adhesion of PET was promoted by PE-CVD of 2-isopropenil-2-oxazoline (2-iox) mixed with Argon at low pressure. The oxazoline derivates are largely used in medical application because of their biocompatibility [19]. Oxazoline polymers, obtained by chemical processes, were already used as additives to improve the adhesion of textile materials; the polymers were used as pre-dip before the RFL application [20]. 2-iox was employed as an organic precursor due to its reactivity. First of all, the oxazoline ring could be fragmentated by the electric potential application during the discharge phase, creating new functional groups. Moreover, the double bond could also react with the radical species that are formed during the discharge. The fragmentation is regulated by the Yasuda Factor, which is defined as the ratio between the Power (W) and the flow. At high values of power, the fragmentation is high, which means that the coating structure is different from the monomer (monomer defect) [21,22]. The aim of the work is to find the plasma coating that guarantees the adhesion between a PET monofilament and the rubber compound. Two different regimes were studied to find the best PE-CVD parameters: a continuous regime and a pulsed regime. In the continuous regime, the variable was the power (W), while in the pulsed regime, the discharge is ruled by the duty cycle, which is defined as the alternation of plasma phase on (fragmentation of 2-iox) and plasma phase off (retention of 2-iox) [21]. The work was divided in two parts; in the first part, the PPOLs were deposited on PET sheets in order to study the surface characteristics and the chemical composition. The wettability was studied, in terms contact angle measurement and thickness, using a profilometer. The chemical composition of the surface was investigated using ATR-IR (Attenuated Total Reflectance Infrared) and XPS (X-ray photoelectron spectroscopy) instruments; the adhesion was studied by Peel Test performed with a tensile machine using the RFL dip adhesion as reference. The characterizations performed on the PET sheets made it possible to optimize the PE-CVD parameters to obtain the coating that guarantees the best adhesion between the fiber reinforcements and the matrix. In the second phase, the optimized PE-CVD of 2-iox was performed on the PET monofilament and the adhesion was studied by the CRA (Cord Rubber Adhesion) test. The degree of coverage was estimated by Optical Microscopy (OM), while TEM (Transmission Electron Microscopy) images were collected to evaluate the thickness and the uniformity of the coating.

## 2. Materials and Methods

### 2.1. Materials

The precursor of the PE-CVD was 2-isopropenyl-2oxazoline (≥99%) and was supplied by Sigma-Aldrich (St. Louis, MO, USA). Polyethylene Terephthalate (PET) was prepared in two different forms: in the first part of the experiment, PET pellets (MA.RE s.p.a, Brescia, Italy) were used to prepare sheets (127.0 mm × 12.5 mm × 3.0 mm). In the second part, the same PET pellets were used to prepare monofilaments (MA.RE s.p.a) with a diameter of

0.40 mm. For the thickness measurement, silicon wafers (10 mm $\times$ 10 mm) were used as substrate, while for the ATR-IR spectroscopy investigations, the coatings were deposited on aluminum foils.

### 2.2. Plasma Reactor and Parameters

A cold plasma was created in a laboratory scale-reactor, composed of a stainless-steel chamber (Plasma Prometeo Labs, Milano, Italy) and equipped with parallel plates as electrodes. The upper electrode is connected to an antenna that provides the radiofrequency, 13.56 MHz, supplied from a generator [23,24]. Before the application of the plasma discharge, the internal pressure of the chamber was reduced by two pumps—a rotary pump and a turbomolecular pump—to reach $10^{-3}$ Pa. A nitrogen liquid trap was placed to protect the rotary pump from the 2-iox gas, which could contaminate the oil [25]. The 2-iox enters in the stainless chamber thanks to a micrometric valve. The flow of the organic precursor was estimated according to the weights of the box containing the organic precursor before and after the treatment using Equation (1):

$$F = \left(\frac{g}{PM}\right)\frac{22400 \text{ cm}^3}{t} \tag{1}$$

where $F$ is Flow, $t$ is the time of treatment, $g$ is the difference of weights in grams and $PM$ is the molecular weight of the 2-iox.

To increase the radical species formation on the substrate, which leads to the growth of PPOL, Argon was inserted in the chamber by a EL-Flow series F 201C by Bronkhorst (Ruurlo, Netherlands). In the continuous regime, a pre-activation by Argon plasma discharge was performed (100 W, 60 s), while in the pulsed regime, the plasma was a mixture between 2-iox and Argon. As reported in previous work, the flows of 2-iox (3.3 standard cubic centimeters per minute, sccm) and Argon (2.3 mln/min) were kept constant, and the time treatment was also fixed at 20 min [21]. For the plasma generated in the continuous regime, the variable was the Power (W) supplied to the system. The Power was tested between 20 and 175 W. For the pulsed plasma, the variable was the duty cycle (d.c.) tested between 10% and 70%; the Power supplied was kept constant at 175 W [15].

### 2.3. Characterizations Techniques

To study the relation between the Power/d.c. with thickness, a Bruker Dektak XT profilometer (Billerica, MA, USA) was used. For the characterization analyses, the PPOL was deposited on silicon wafers and the height of the coating was measured by means of the application of a tape mask. The wettability of the coating was investigated on PET sheets by measuring the contact angle (c.a.), performed with the Dataphysics OCA20 (Filderstadt, Germany) instrument in the air at room temperature. The degree of the wettability of a surface can be defined by the Young relation, which describes the value of the c.a. between the solid substrate and a drop of water. The surface can be hydrophobic (c.a. $> 90°$) or hydrophilic (c.a. $< 90°$) [24,26,27]. For this measurement, a drop of water (3 $\mu$L) was deposited on the surface of the coating and the c.a. was estimated as average by measuring the c.a. at 5 different positions.

To study the chemical structure of the coating and to investigate the fragmentation of the 2-iox, Attenuated Total Reflectance Fourier-Transform Infrared spectroscopy (ATR-FTIR) was used. With this technique, it is possible to obtain information at the molecular level [28]. The ATR-FTIR spectra were acquired with a Nicolet iS10 spectrometer (Thermo Scientific, Waltham, MA, USA) equipped with an iTR Smart device (total scan 32, range 4000–800 cm$^{-1}$, resolution 2 cm$^{-1}$). For this analysis, the PPOL was deposited on aluminum foils. X-ray Photoelectron Spectroscopy (XPS) was used to investigate, in greater detail, the fragmentation at two different d.c. conditions: 10% and 50% [29]. The analysis was performed at ICMATE-CNR of Padova (Italy), using a Perkin Elmer $\Phi$ 5600-ci spectrometer (Waltham, MA, USA), and detailed scans were collected for the C1s, O1s and N1s. To measure the thickness of the PPOL on the PET monofilament and to evaluate the unifor-

mity of coating, transmission electron microscopy (TEM) Zeiss LEO 912ab (Oberkochen, Germany) images were acquired. The analysis was performed on the PET monofilament covered by the PPOL, which guaranteed the best adhesion, in comparison with the PET monofilament treated with RFL dip.

### 2.4. Adhesion Studies

The adhesion characterization was performed by different tests depending on the PET shape: sheet or multifilament. The adhesion of the coatings deposited on the PET sheets was evaluated by the measurement of peel strength, which is defined as the force required to progressively separate a rigid member from a flexible member [30]. In the case of the 2-iox coating used as pre-dip, the application of the RFL was made in Sicrem labs at Pizzighettone (Cremona, Italy). For the peel test, the treated sheets were vulcanized with a sheet of rubber compound (30% poly 1,3-butadiene and 70% synthetic rubber, Pirelli tire spa, Milano, Italy) at 8 bar 151 °C for 40 min. The samples were vulcanized immediately after the plasma treatment because the high reactivity of the surface decreases in time for the oxidation processes during the exposure to air [24]. Three samples for each coating were prepared for the peel test. The adhesion on the PET monofilament was investigated by CRA (Cord Rubber Adhesion) test, which measures the extraction force. The monofilaments were vulcanized with the same sheet of compound and condition used for the peel test. To estimate the adhesion, three samples were prepared for each coating. After the CRA test, it was possible to evaluate the adhesion even with the degree of coverage. The coverage was defined as the residual part of the rubber that remains on the reinforcement after the CRA test and the investigation was performed by optical microscopy (Zeiss, Oberkochen, Germany). The degree of coverage is between 0 (no adhesion) and 5 (total adhesion). Both adhesion tests were performed in Pirelli labs (Pirelli tire spa, Milano, Italy) with a Zwick Z10 dynamometer (ZwickRoell, Genova, Italy).

## 3. Results and Discussion

### 3.1. Thickness

The first characterization involved the evaluation of the thickness of the film deposition at different powers; the analysis was performed at the Università degli studi di Milano Bicocca (Milano, Italy). To measure the thickness, different PPOLs were deposited on silicon wafers and the height of the deposition was investigated. In Tables 1 and 2, thickness values (nm) for coatings obtained in the continuous and the pulsed regime are reported. The gas flow and time treatment were kept constant (20 min) in the continuous mode, and, in the pulsed regime, the power was also fixed.

**Table 1.** Thickness of continuous PPOLs; the values were collected at different condition of power (W) in the continuous regime. For each plasma coating, three samples were prepared; the values of the thickness are the average of three measurements.

| Power (W) | Thickness (nm) |
|-----------|----------------|
| 20  | $235 \pm 20$ |
| 50  | $357 \pm 18$ |
| 90  | $364 \pm 21$ |
| 130 | $503 \pm 25$ |
| 175 | $448 \pm 18$ |

**Table 2.** Thickness of pulsed PPOLs, the values were collected at different d.c. conditions. For each plasma coating, three samples were prepared; the values of the thickness are the average of three measurements.

| Duty Cycle (%) | Effective Power (W) (W × d.c.) | Thickness (nm) |
|---|---|---|
| 10% | 18 | 202 ± 10 |
| 30% | 53 | 334 ± 16 |
| 50% | 88 | 417 ± 21 |
| 70% | 123 | 426 ± 25 |
| 100% | 175 | 448 ± 24 |

W × d.c. is power multiple by the duty cycle.

The data indicated that the thickness increased with the Yasuda factor (Ø2-iox = 3.3 sccm): the height of the deposition is usually related to the fragmentation; at high power, the number of reactive species is higher, allowing the higher depositions [16]. Coatings obtained in the pulsed regime had lower thickness, due to lower precursor fragmentation.

*3.2. Wettability*

The wettability was measured for the different coatings deposited at different conditions on the PET sheets. For PPOLs obtained in the continuous regime, the data are related to an Argon pre-activation followed by a PE-CVD of 2-iox at different powers, in the range 20–175 W. The values are reported in Figure 1 and compared with the values of the untreated PET and with the values of the monomer. The contact angle (c.a.) of the 2-iox is an important reference point in order to understand the level of the fragmentation. The c.a. measured highlighted that the coatings, at any power tested (except at 175 W), were characterized by higher hydrophilicity in comparison to the untreated PET; the values are between 70° and 80° (<90°). Low contact angles in the continuous regime indicate that the fragmentation happens even at low power (20 W) because they are significantly higher than the c.a. values collected for the pure monomer [21]. To evaluate the stability of the coating, the contact angle was measured again after five days and after water rinsing. The aging did not affect the wettability, and the continuous regime PPOL deposits were considered stable.

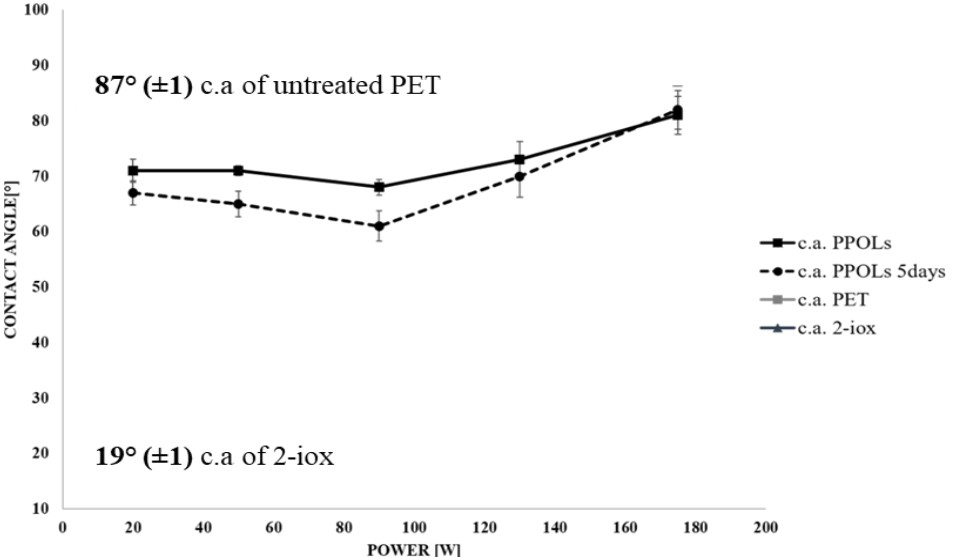

**Figure 1.** c.a. in the continuous regime at different powers; the values were measured on PET sheets. The values of the contact angle are the average of 5 values collected for each sample.

The contact angle values were also measured on the coatings obtained for different duty cycle conditions from 10% to 100%, while the gas flow and the time treatments were

kept constant. The values collected in the pulsed regime are reported in Figure 2. The PPOLs had higher hydrophilicity in comparison to the untreated PET; the c.a. values are between 60° and 68°. An exception was observed for the coating obtained at 10% d.c. (18 W $W_{eff}$); this showed high hydrophilicity with a c.a. similar to the value of the monomer. It is possible to assume that, at low duty cycles, the fragmentation is minimal; indeed, the wettability of the PPOL at 10% d.c. is similar to that of the 2-iox monomer. As reported for the continuous regime, the ageing did not affect the wettability except for the 10% d.c. It was interesting, in fact, to note that the contact angle for the 10% d.c. PPOL changed because of the hydrolysis of the retained oxazoline rings after the rinsing in water [21].

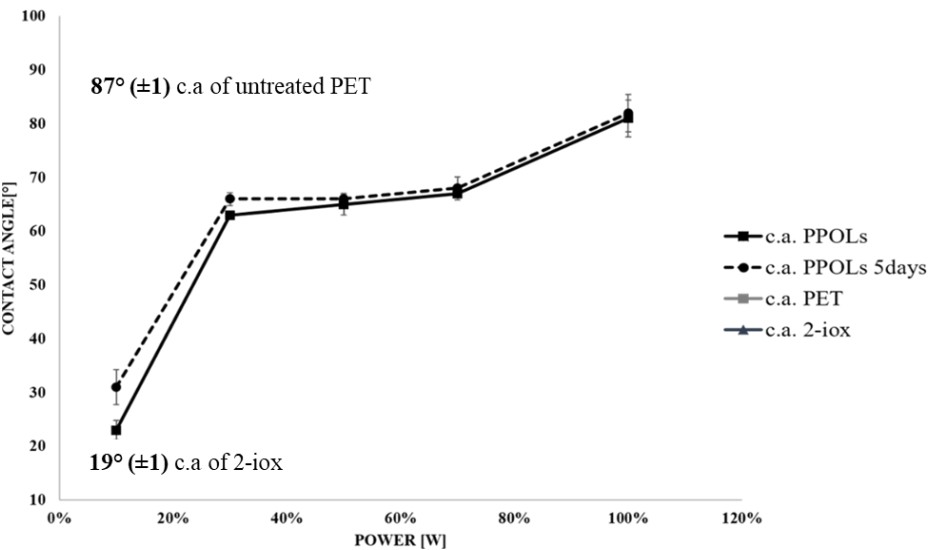

**Figure 2.** c.a. in the pulsed regime at different powers; the values were measured on PET sheets. The values of the contact angle are the average of 5 values collected for each sample.

### 3.3. Chemical Characterizations

A chemical characterization of the different 2-iox plasma coatings was performed by FTIR spectroscopy. The analyses were exploited to investigate the chemical composition of the PPOLs. The IR spectrum of the monomer (2-isopropenyl-2-oxazoline) was collected to identify the main characteristic peaks of the molecule and compared with the PPOLs spectra obtained with 10% d.c. (minimal fragmentation) and 50% d.c. (fragmentation). The overlapped spectra are reported in Figure 3. Concerning the PPOLs obtained at low d.c, the characteristic peaks of the monomer were detected, except for the signal of the double bond C=C of the isopropenyl group which reacted with the radical species formed on the surface by the plasma discharge [19]. At 1662 cm$^{-1}$, it is possible to identify the sharp peak relative to the C=N, at around 3000 cm$^{-1}$ appears the peaks characteristic of the CH$_x$ stretching. Another characteristic peak of the oxazoline ring is recognizable at 1182 cm$^{-1}$; this is relative to the bending of the C–O [26]. The investigation performed by ATR-IR confirmed the results obtained with the wettability studies; at 10% d.c. plasma, the oxazoline rings were partially retained. With the increase in the d.c., the profile of the spectra changed. In the PPOL obtained at 50% d.c. spectrum, the peak relative to the C–O bond at 1182 cm$^{-1}$ was not detected, while the C=N peak at 1662 cm$^{-1}$ was broader with an important shoulder at high wavenumbers, which was probably related to the formation of carboxylic and carbonyl groups from the oxazoline ring opening. In the spectra of the PPOL obtained at 50% d.c. (Figure 3), a small shift occurs in the zone of the peak of the double bond carbon nitrogen; the chemical neighborhood is changed due to the fragmentation. The degree of the fragmentation is higher than that of the 10% d.c. discharge but some peaks of the monomer are recognizable, such as the CH$x$ and the double bond C=N. The area of the spectra between 1950 and 2350 cm$^{-1}$ is characteristic of

the isocyanate (–N=C=O) peaks; in this zone, small peaks are present due to the higher percentage of plasma phase on fragmentation [21–26]. Figure 4 shows the spectra of the coating obtained in the continuous regime; the characteristic peak of the 2-iox (C–O) is not recognizable, and the shift in the zone of the C=N indicates that the chemical structure was changed due to the high fragmentation. In the isocyanate (–N=C=O) area, small peaks are identified [31].

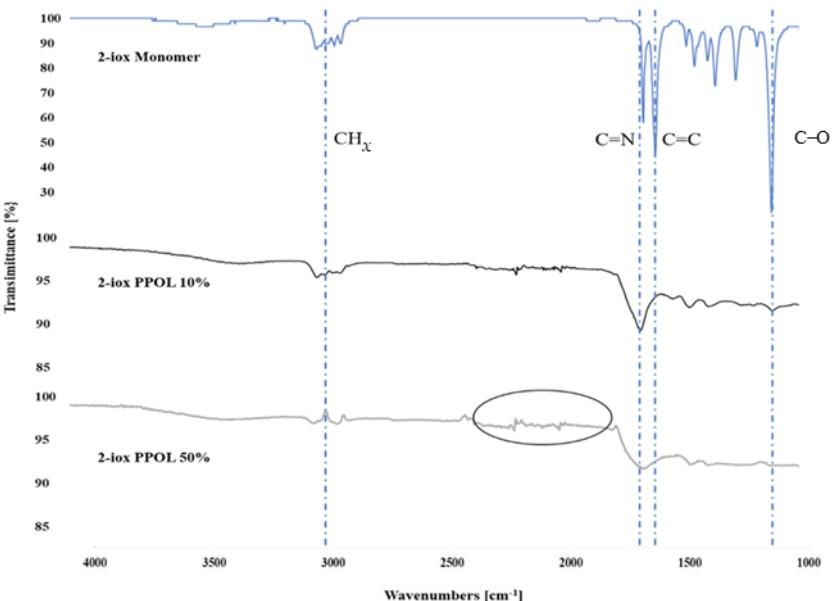

**Figure 3.** ATR-FTIR spectra of the 2-iox, 10% PPOL and 50% PPOL. The peak at 3000 cm$^{-1}$ is relative to CH$_x$; at 1662 cm$^{-1}$, the C=N peak is identifiable; at 1630 cm$^{-1}$, the signal is relative to the C=C; and at 1182 cm$^{-1}$, the peak of C–O is identifiable.

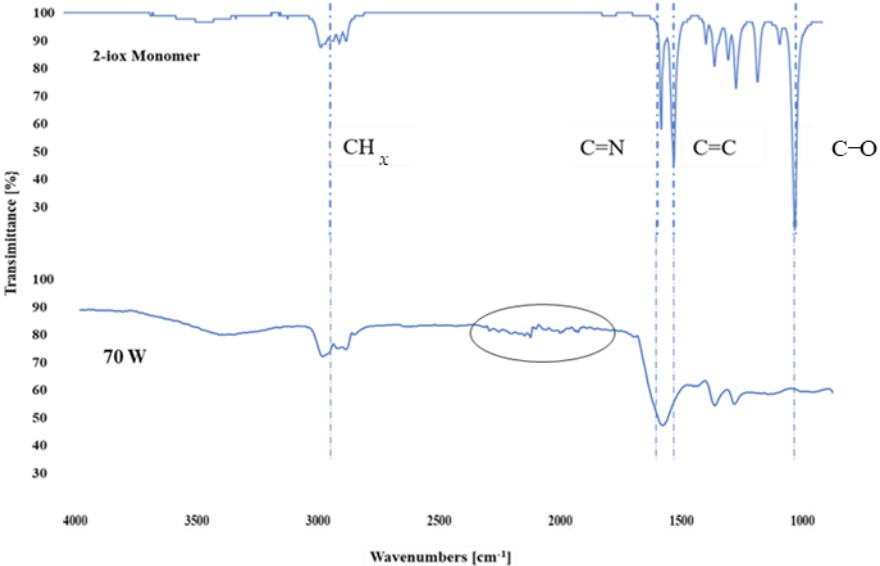

**Figure 4.** ATR-FTIR spectra of the monomer and 70 W PPOL. The peak at 3000 cm$^{-1}$ is relative to CH$x$; at 1662 cm$^{-1}$, the C=N peak is identifiable; at 1630 cm$^{-1}$, the signal is relative to the C=C. The peak of C–O is no longer identifiable.

A preliminary XPS analysis was performed at ICMATE-CNR (Padova, Italy). In particular, three PET sheets were characterized: the untreated and two PPOLs deposited at different d.c. conditions (Table 3). The difference between the coatings was not so

clear because of "the noise" of the substrate where the coating was deposited. The most important indication is the increase in the fragmentation with the increase in the power: the percentage of the carbon increases with the power, the percentage of oxygen and nitrogen decreases with the power. The presence of the nitrogen (characteristic of the 2-iox) can be noticed in the treated sheets. The rupture of the oxazoline ring at 50% d.c. could be confirmed by the CO (characteristic of the 2-iox) contribution. The CO contribution was evaluated with the ratio CO/C% (Table 4). The percentage calculation was performed on 10% d.c. PPOL and 50% d.c. PPOL: the CO/C% seems to decrease with the increment of the fragmentation [31]. However, these results are preliminary; more detailed analysis will be performed on silicon wafer in order to avoid interference with PET.

**Table 3.** Atomic composition of the samples determined by XPS. The composition is expressed in percentage, the analysis was performed on PET, 10% PPOLs, 50% PPOLs and 175 W PPOLs. The experimental uncertainty of the reported atomic composition values does not exceed ± 5%.

| Element | **C** | | | **O** | | **N** |
|---|---|---|---|---|---|---|
| Sample | C–C C=C | C–O | O–C=O | C–O–C | C=O | – |
| Pristine | | 83.4% | | | 16.6% | – |
| | 74.1% | 5.7% | 3.6% | 8.4% | 8.2% | – |
| 10% | | 73.9% | | | 12.5% | 13.6% |
| | 47.1% | 19.2% | 7.6% | 6.3% | 6.2% | |
| 50% | | 76.7% | | | 10.7% | 12.6% |
| | 53.0% | 17.3% | 6.4% | 5.3% | 5.4% | |
| 175 W | | 77.9% | | | 10.2% | 10.9% |
| | 55.1% | 18.7% | 4.1% | 5.1% | 5.1% | |

**Table 4.** Ratio CO/C for different d.c. conditions. The ratio is calculated for the PPOLs obtained at 10% d.c. and at 50% d.c.

| PPOL | CO/C% |
|---|---|
| 10% d.c. | 26% |
| 50% d.c. | 22% |

### 3.4. Adhesion Characterization

Peel strength between the coatings, deposited on the PET sheets, and a rubber model compound was measured with a dynamometer. The PPOLs were tested both as adhesive and as pre-dip to activate the PET for the RFL treatment. Table 5 reports the values of the average peel force. The coatings with a high degree of fragmentation (on continuous plasma) reacted better as pre-dip. On the other hand, the pulsed plasma polymer coatings reacted better as adhesives. The 10% d.c. PPOL deposit did not promote adhesion, and the oxazoline ring seemed not to be reactive with the rubber or with RFL; the 50% PPOL promoted the adhesion directly, without the RFL dip. A PET sheet treated by the classical adhesion method was used as a reference: the adhesion of the 50% PPOL is comparable with the one obtained with the classical method (epoxy-isocyanate bath + RFL).

The plasma parameters that gave the best adhesive coating were used to treat the PET monofilament which is the target material. Low Pressure Plasma (LPP) treatment (W = 175, d.c. = 50%, t.t = 20 min, Ø2-iox = 3.3 sccm; ØAr = 2.3 mln/min) was performed on the PET monofilament (diameter 0.40 mm), and the sample was vulcanized with a rubber sheet. The PPOL adhesion was tested by the CRA (Cord Rubber Adhesion) test, which measures the extraction force, while the degree of coverage was evaluated by optical microscopy. The extraction force and the coverage degree of PET were improved after the plasma treatment; nevertheless, the PPOL looked scraped, and the coating was not well stabilized on the PET

monofilament (Figure 5). To improve the stability between the plasma coating and the PET monofilament, and therefore the adhesion with the rubber, a plasma activation was performed on the samples before the PE-CVD with 2-iox. The plasma activation was made before the PE-CVD, in the same reactor, using Argon in the continuous regime for 120 s (100 W). This plasma process, performed using an inert gas, makes it possible to create radical species on the surface, which make the material more reactive [21]. The PPOLs' extraction force and the degree of coverage were then compared with the adhesion results obtained on PET mono treated with epoxy bath + RFL (classical treatment). For each PPOL or RFL dip, three samples were prepared to evaluate the adhesion. The values collected in Table 6 are the averages of three measurements. The images collected by the optical microscopy (Figure 5) show the success of the activation; the back residual represents the part of the rubber that remained on the monofilament after the CRA test, the residual increases with the pre-activations. It was possible to observe that the residual of rubber completely covered the monofilament (pre-activated with Argon Plasma) after the CRA test. Moreover, the extraction force of the activated PPOL was comparable (slightly lower) with the reference (PET + epoxy +RFL). In conclusion, the degree of coverage was higher for the plasma-treated cord.

**Table 5.** Peel test results for the PPOLs obtained at 50, 90 W and at 10% and 50% d.c. In the table, V is used when a treatment is performed, and X is used when no treatment is performed.

| Plasma Regime | PPOL | PE-CVD (2-iox) | Epoxy + RFL | Avg. Peel Force | De.st. |
|---|---|---|---|---|---|
| Continuous | 50 W | V | V | 36.9 N | 4.2 |
| Continuous | 50 W | V | X | 22.9 N | 1.9 |
| Continuous | 90 W | V | V | 33.6 N | 5.8 |
| Continuous | 90 W | V | X | 23.1 N | 1.7 |
| Pulsed | 10% | V | X | 19.2 N | 2.3 |
| Pulsed | 10% | V | V | 14.8 N | 3.0 |
| Pulsed | 50% | V | X | 55.3 N | 5.9 |
| Pulsed | 50% | V | V | 15.3 N | 2.3 |
| – | EPOXY+ISOCYANATE | | V | 52.9 N | 4.7 |
| PET | UNTREATED | | X | 11.3 N | 2.9 |

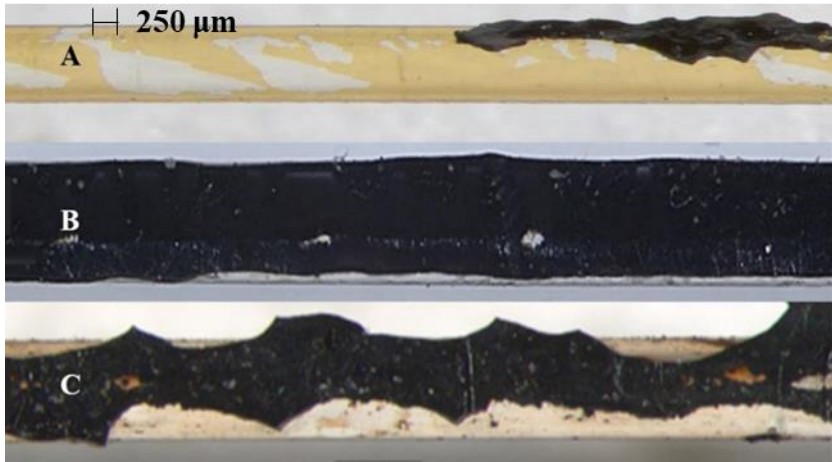

**Figure 5.** Coverage images of 50% PPOL on PET monofilament without activation (250 μm) (**A**), with plasma activation (**B**) and with Epoxy + RFL activation (**C**).

**Table 6.** CRA test results for the PPOLs obtained at 50% d.c. that differ for the Ar activation. The adhesion is expressed by the extraction force and the coverage. The adhesion of the PPOLs is compared with the adhesion of the RFL system and with the untreated PET. In the table, V is used when a treatment is performed, and X is used when no treatment is performed.

| Covering | Ar Activation | PE-CVD | Epoxy + RFL | Avg. Peel Force | De.st. | Coverage |
|---|---|---|---|---|---|---|
| 50% PPOL | X | V | X | 6.3 N | 0.5 | 1 |
| 50% PPOL | V | V | X | 14.3 N | 1.7 | 4 |
| CLASSICAL | EPOXY+ISOCYANATE | | V | 19.7 N | 1.3 | 3 |
| PET | UNTREATED | | X | 4.2 N | 0.2 | 0 |

TEM images were collected (Figure 6) for further examination of the two adhesives, the 2-iox PPOL and the RFL. The two adhesives differ in terms of the thickness and uniformity of the covering; in fact, the measure of the height of the PPOL gave a constant value (1.1 μm) and the height of the RFL dip was included in a range between 1.7 and 2.8 μm.

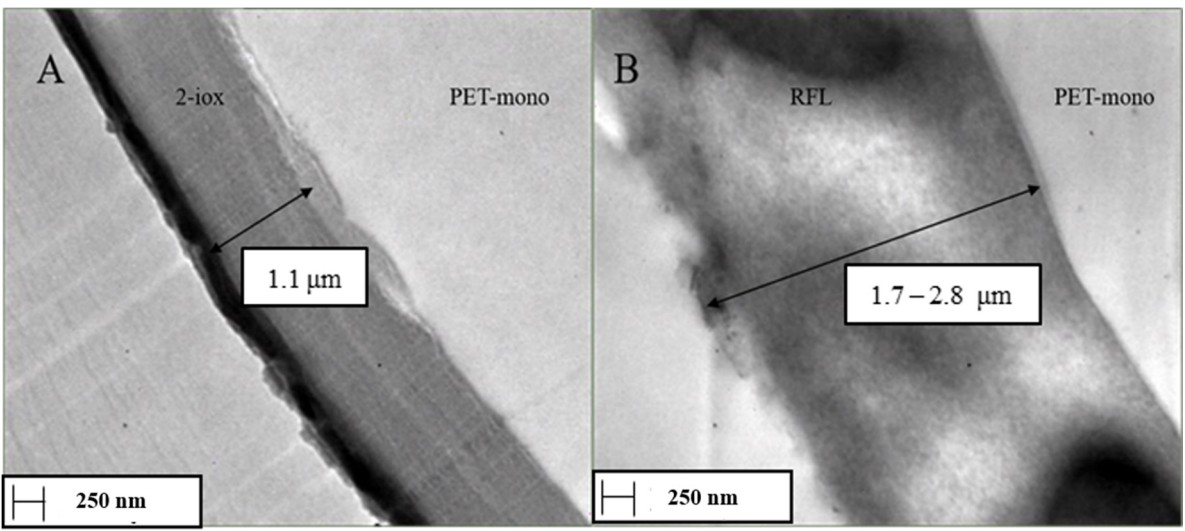

**Figure 6.** TEM images of activated 2-iox PPOL (**A**) and Epoxy + RFL adhesive (**B**). Thickness is 1.1. μm for the PPOLs and between 1.7 and 2.8 μm for the Epoxy + RFL.

## 4. Conclusions

The aim of the work was to find an environmentally friendly method, as an alternative to the classical method (epoxy bath + RFL), to guarantee the adhesion between PET monofilament and rubber compounds. Two different regimes of plasma (continuous and pulsed) were used to perform a PE-CVD of 2-isopropenyl-2-oxazoline. The PPOLs were deposited at different powers and duty cycles, while the treatment time and flow of Argon and 2-iox were kept constant. The surface characterizations were carried out on PET sheets. The c.a. measurement highlighted how the fragmentation happens even at low power (20 W) but does not occur at low duty cycle (10% d.c.). In this condition, the oxazoline rings were retained. The characterizations with ATR-IR and XPS confirmed the c.a. analyses, indicating that, in the 10% d.c. PPOL, the characteristic peaks of the monomer are recognizable. The PPOLs were tested, both as adhesives and pre-dips for the RFL, by a peel test. The result of the adhesion test highlighted how the PPOL, obtained in the pulsed regime at 50% duty cycle, has comparable peel force to that of the chemical dip. The PE-CVD in the pulsed regime was probably more controlled than the treatment in the continuous regime, where the monomer is highly fragmented by the plasma discharge [16–21]. This could facilitate the development of a network between the chains of the plasma polymer and the rubber during the vulcanization. The plasma parameters that

gave the best adhesion were used to perform the PE-CVD (50% d.c. 175 W) of 2-iox on the PET monofilament. The CRA test and the evaluation of the coverage degree pointed out the necessity to pre-activate the PPOL in order to improve the stability between the PET and the coating. The extraction force and the coverage degree of monofilament treated by PE-CVD with 2-iox, pre-activated with argon plasma, have comparable values with those obtained with the RFL, pre-activated with isocyanate-epoxy bath. Those adhesion properties were obtained with an environmentally friendly plasma procedure with low doses of reagents and without any solvent. The work is intended to be a preliminary study to evaluate the monomer and the process under controlled conditions (vacuum). In the future, the same system will be studied at atmospheric pressure.

## 5. Patents

Process of manufacturing a reinforcement component for a tire and related tire process (C.Gaifami, B.Rampana, P.Caracino, S.Agresti, C.Riccardi) (2020P00011IT)

**Author Contributions:** Investigation, C.M.G.; Supervision, C.R.; Validation, S.Z. and L.Z. All authors have read and agreed to the published version of the manuscript.

**Funding:** This research received no external funding.

**Institutional Review Board Statement:** Not applicable.

**Informed Consent Statement:** Not applicable.

**Data Availability Statement:** The data that support the findings of this study are available from the corresponding author upon reasonable request.

**Conflicts of Interest:** The authors declare no conflict of interest.

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
