# Peer review of "Plasma Enhanced-Chemical Vapor Deposition of 2-Isopropenyl-2-Oxazoline to Promote the Adhesion between a Polyethylene Terephthalate Monofilament and the Rubber in a Tire"

_coatings, doi:10.3390/coatings11060708_

Round 1

Reviewer 1 Report

This manuscript describes using a plasma-enhanced chemical vapor deposition to deposit organic thin films on a polyethylene terephthalate monofilament to increase adhesion with the rubber in a tire. The goal is to replace the chemical dipping method with a more ‘green’ method in terms of the environmental waste that is produced. The wettability was determined by the contact angle approach and the thickness by a profilometer. The results were also characterized by infrared and X-ray photoelectron spectroscopies. The adhesion of the polyethylene terephthalate sheets was measured by the peel test.

I have a few comments that need to be addressed. A key issue that the authors need to address is how practical will this process be for a large-scale industry such as tire manufacturing as it has to be done in vacuum. Also, what are the energy requirements for the plasma and the generation of the vacuum. This will affect the ‘greenness’ of the process.

The work itself is overall of good quality.

Equation (1) is missing.

Line 135 what is level 28? Is this a typo for a reference?

The authors do an excellent job on including error bars in tables and figures.

Give some references for the IR assignments.

Line 234. The -1 in cm-1 should be a superscript.

Figure 3. top spectrum, it should be C-O not C=O.  see figure 4 for correct labeling.

The references do not appear to be complete in terms of article titles or complete references. as an example, I would have no idea where to find reference 9. What is it?

The work is interesting in terms of the new approach and the work is appropriate for Coatings after the corrections are made.

Author Response

Dear Reviewer,

thank you for your comments.

At the line n° 372 i added a comment about your first ossevation about the "greeness" of a vacuum process. This work could be considered as a preliminary study for the further studies in atmospheric condition. I spoke about green technology because of the low doses of chemicals which are involved in plasma treatment.

1) the equation 1 was added

2) Yes it was a reference, i had a problem during the conversion of the file. Thank you

3) The reference for the IR analysis ais the reference 31

4) I corrected line 234

5) i corrected figure 3

6) I implemented te bibliograhy and i also added the iperlink

Reviewer 2 Report

The manuscript reports plasma-enhanced chemical vapor-deposition of an organic thin film on polyethylene terephthalate monofilament to increase its adhesion with the rubber compound in a tire.

The work is interesting representing a green and environmentally friendly method, using 2-isoprepenyl-2-oxazoline (2-iox) as precursor. The work should be published after minor revisions.

  1. The authors should to place the isocyanate in the fig 3 in the circled surface and also in fig 4.

Author Response

Dear reviewer,

thank you for your comments.

I corrected the figure 4.

Reviewer 3 Report

This article reports the surface modification of the PET through PE-CVD processes. The characterization of the plasma polymer film and the adhesion properties is important for future environmental friendly methods.

Before further stages, authors should address some considerations as following.

1) I feel that English editing is needed. For example, at line 48-50, the original text is below:

“Thanks to this environmentally friendly techniques, is possible to change the morphology of the surface or to deposit a thin film on different substrates using limited doses of reagents”

Before “is possible to change..”, inserting “it” might be one possible route.

I also found same points at line 129, and line 139.

2) Especially at “Materials and Methods” and “Results and Discussion” section, authors should confirm the number for “citation” or real description.

One example is: “..supplied from a generator 23-24.” at line 103-104.

3) I cannot find equation 1.

4) At Figure 1 and the related main text: 

The contact angles of PPOLs (in Figure 1) are not included between 60° and 70°, as written at line 196 and 197. It seems that five days aging sample might be consistent. Also, the contact angle data description about the untreated PET and 2-iox monomer might be misleading, because there are plots at same plasma power even without plasma treatment.

If possible, please modify this Figure 1 and also Figure 2.

5) At Table 5 and 6, author use “X” for “apply this process”, and “(blank)” for “not apply”. Other symbol might be better to understand.

6) At Figure 5 ‘coverage image”, adding some description in the figure is better. I think that black zone is “rubber”.

7) At line 341, “Figure 5” should be “Figure 6”, because the coverage image is “Figure 5”.

Please confirm the consistency with Figure, Table and main text, carefully.

Author Response

Dear reviewer,

thank you for your comments.

1) I revised and corrected the text, thank you for your observation

2) I corrected the reference format and i added a iperlink

3) The equation 1 was added

4) I corrected the line 196-197 and i also corrected the figure 1 and 2

5) I corrected the tables. I used "V" when a treatment was performed and "X" when no treatments were performed on the sample

6) From the line 318 to the line 323 i added some comments about the figure 5

7) I corrected the number of the figure and i checked the consistency of the figure, table and text. 

Reviewer 4 Report

The authors have reported a “green” technique by plasma polymerisation of 2-isopropenyl-2-oxazoline on PET to enhance the adhesion with rubber. The plasma-mediated polymer deposition provides a simple and effective way to produce a thin adhesive film on PET without using those environmentally concerning chemicals. The results from peeling test confirmed the adhesion between PET and rubber is comparable to that by the classical method with RFL. While the results are interesting, there are some details that should be clarified by the authors before the manuscript can be accepted for publication.

  1. it is not very clear what thickness of the plasma polymer was used in the final adhesion. is it 1.1 um shown in the TEM graph? why is this thickness chosen? Have the authors looked at how the thickness of polymer coating affects the adhesion between PET and rubber? In Table 5, when the peel test results under different plasma polymerisation conditions were compared, has the thickness variation (shown in Table 1&2) been taken into account? It is important to have a full profile of the polymer coating for the optimal adhesion result.
  2. Have the authors examined the stability of the polymer coating over time? Is the attachment between polymer coated PET and rubber immediately after the plasma polymerisation process? Because plasma polymer with radicals and reactive moieties would be oxidised over time. As indicated by the results, higher fragmentation (more reactive/functional moieties on the plasma polymer) leads to better adhesion. It is worth indicating if the plasma polymer has been adhesive in a long term or the vulcanisation happens straight after plasma treatment.
  3. The XPS results shown in the manuscript are not conclusive. As mentioned by the authors, the PET substrate itself has C-O, O=C-O species that overlay with 2-iox polymer, making the comparison difficult. Moreover, the binding energy for C-O and C-N is very close. It would be hard to distinguish the two species when fitting, making the ratio CO/C in table 4 questionable. It is recommended that the authors should run the XPS scans with plasma polymer on silicon wafers to avoid interference from PET.

Author Response

Dear reviewer,

thank you for you comments.

1) In the first part of the work the thickness was studied in order to evaluate the deposition rate, in the second part the thickness was measured to compare the rfl coating and plasma coating, the measure of the thickness was presented also in order to point out the uniformity of the coating. In general the adhesion is dominated by the chemical characteristic of the coating, rather then of the thickness in the range of the order of 1 um. (reference n°16 and reference n°21).

2) Authors examined the stability after 5 days and after a the rinsing in water, data are reported in figures 1and 2. The attachment between polymer coated PET and rubber is performed immediately after the plasma deposition process. We added a paragraph in order to specify that the vulcanization must be done after the plasma treatment because of the high reactivity of the surface (reference 24) line 157 to 159.

3) we agree with the referee that the results are not conclusive. We change the paragraph 260-275, including also the sentence “However these results are preliminary more detailed analysis will be further performed also on silicon wafer in order to avoid interference with PET.”

Thank you for your kindness

Best regards